# Study on the Structure and Skin Moisturizing Properties of Hyaluronic Acid Viscose Fiber Seamless Knitted Fabric for Autumn and Winter

**DOI:** 10.3390/ma15051806

**Published:** 2022-02-28

**Authors:** Qiuyu Wang, Jialiang Lu, Zimin Jin, Kun Chen, Mingtao Zhao, Yuqiang Sun

**Affiliations:** 1College of Textile Science and Engineering, Zhejiang Sci-Tech University, Hangzhou 310018, China; w118572@163.com (Q.W.); kivenjin@163.com (Z.J.); 17826858273@163.com (K.C.); 2Zhejiang Bangjie Holding Group Co., Ltd., Yiwu 322009, China; mtzhao@bangjie.cn; 3College of Life Sciences and Medicine, Zhejiang Sci-Tech University, Hangzhou 310018, China; sunyuqiang@zstu.edu.cn

**Keywords:** hyaluronic acid, seamless knitted fabric, skin moisture content, trans epidermal water loss

## Abstract

In autumn and winter, the climate is dry and human skin tends to become dry as a result. The application of hyaluronic acid to a fabric has a certain moisturizing effect, which can further improve the wearing comfort of the fabric. In this study, three knitted fabric structures including weft plain stitch, 1 + 1 mock rib, and 1 + 3 mock rib were designed. The face yarn adopted hyaluronic acid viscose fiber, graphene viscose fiber, nylon, and viscose with different interweaving ratios, and the inner yarn adopted nylon/spandex-coated wire. A total of 18 sample knitted fabrics were woven according to the experimental method of the full test. The effects of the fabric structure and fabric raw materials on skin moisturizing properties were studied and analyzed by testing the skin water content and trans-epidermal water loss before and after coating the fabric on the human skin. The results show that the most significant factor affecting the skin moisture content is the raw material used in the fabric. When the interweaving ratio of hyaluronic acid viscose fiber in the fabric decreases, the moisturizing performance of the fabric on human skin is weakened. The second is the fabric structure. In terms of structure, 1 + 1 mock rib fabric has better moisturizing performance for human skin. When the material adopts hyaluronic acid viscose fiber/graphene viscose fiber (100:0) and the structure adopts 1 + 1 mock rib, the moisturizing effect on human skin is better.

## 1. Introduction

Against the background of the development of functional textiles, hyaluronic acid textiles with moisturizing properties have attracted more and more attention. Hyaluronic acid (HA) is one of the components of human skin, and is the most widely distributed acidic mucopolysaccharide in the human body. It exists in the connective tissue matrix and has a good moisturizing effect. At present, HA is widely used in cosmetics and is known as an ideal natural moisturizing factor [1]. In addition to its moisturizing function, hyaluronic acid also has many important physiological functions and is widely used in medicine, such as lubricating joints and accelerating wound healing. Pavel Suchý et al. prepared three new non-woven fabrics and found that the non-woven fabric prepared by adding hyaluronic acid had a good effect on rapid hemostasis. The fabrics yielded favorable hemostatic activity, bioresorbability, and non-irritability, and had a beneficial effect on tissue repair [2]. Some researchers made sterile hyaluronic acid nanofibers into wound dressings, and then compared them with other types of dressings. It was found that the gauze woven from hyaluronic acid nanofibers had the best healing effect on the wound [3]. HA solution is a typical non-Newtonian fluid with excellent lubricity and viscoelasticity, which makes it widely used in ophthalmology, orthopedics, gynecology, general surgery, and other fields [4]. Hyaluronic acid hydrogel is a promising delivery system for topical applications. Beyond this, HA can also promote local drug absorption and is highly compatible with unstable biomacromolecules [5].

For the research on the application of hyaluronic acid in textiles, scholars from Hong Kong Polytechnic University have shown through experiments that skincare textiles may enhance the regeneration of cells in the human skin structure, so as to improve the state of the skin [6]. Other scholars use the DPPH method to evaluate the anti-free radical ability of antioxidants, and then evaluate the efficacy of skincare textiles [7]. Zuzanna J. Krysiak et al. believed that moisturization is the key in eczema treatment as dry skin triggers inflammation that damages the skin barrier. They combined electrospun hydrophobic polystyrene (PS) and hydrophilic nylon 6 (PA6) with oils to create patches helping to moisturize atopic skin. They showed the great potential of their designed patches through oil release tests on skin and their moisturizing effects were verified [8]. Chaheh et al. have shown that when the protease is worn, many active ingredients will be produced at the contact part between the skin and textiles, resulting in the moisturizing effect of vitamin E on the skin [9]. In the field of the preparation of hyaluronic acid fibers, some scholars have also achieved process innovation in recent years, such as scholars from Zhejiang Sci-Tech University, who studied a silk fabric skincare finishing agent with Acacia senegal and microcapsules made of modified starch as wall material. According to the comparison of the comprehensive properties of the finished fabric, CWS vitamin E microcapsules were optimized, and the suitable silk fabric microcapsule skincare finishing process and formula were obtained [10]. Sergej Karel et al. described fibers based on hyaluronic acid (HA) and their properties under solid-phase peptide synthesis (SPPS) conditions. The fibers were prepared by wet spinning technology and were studied for their stability and mechanical properties under the reaction conditions for direct peptide synthesis or conjugation with a pre-synthesized peptide. The functionality of the modified fabric was confirmed by experiments. Therefore, HA carriers bearing peptides could be advantageously employed in various biomedical applications as repair patches [11].

It can be seen from the above that hyaluronic acid and its derivatives are widely used in the biomedical field. At present, there is relatively little research on hyaluronic acid moisturizing textiles. Hyaluronic acid fiber still has great application prospects in the field of clothing textiles, which is closer to people’s daily lives. In addition, the test methods and evaluation criteria for moisturizing textiles are not perfect, and many aspects still need more research. Seamless knitting technology is not bound by the elasticity of knitwear by suture, and it has attracted more and more attention in the market in recent years. Most textiles directly come into contact with human skin, but human skin is very fragile and sensitive. The skin is prone to be rough, especially in autumn and winter. Additionally, the nerve endings in the skin are more vulnerable to external stimulation due to epidermal shedding, resulting in uncomfortable symptoms such as skin pruritus and dryness [12]. Therefore, it is of great application value to design and develop hyaluronic acid seamless knitted products with excellent moisturizing effects by combining seamless knitting technology and hyaluronic acid viscose fiber. Aiming at the moisturizing function of hyaluronic acid viscose fiber on human skin, this paper analyzes the influence of the structure and raw materials of hyaluronic acid viscose fiber seamless knitted fabric on the moisturizing performance on human skin. The reference theoretical basis and test method for moisturizing seamless knitted clothing containing hyaluronic acid viscose fiber are put forward.

## 2. Materials and Methods

### 2.1. Sample

This paper mainly studies the moisturizing performance of hyaluronic acid viscose fiber seamless knitted fabric on human skin; at the same time, graphene viscose fiber is added to improve its warmth retention and comfort in autumn and winter. Hyaluronic acid viscose fiber, graphene viscose fiber, nylon, and viscose fiber are selected as the veil yarn. The graphene viscose fiber is a cotton staple fiber. In order to increase the elasticity, resilience, and wearing comfort of the fabric, nylon and nylon/spandex-coated yarn are selected as the lining yarn. The front loop is the veil and the back loop is the lining. Among the yarn raw materials, graphene viscose fiber is a short fiber whose staple length is 38 mm, while other yarns are filaments. The breaking strength of graphene viscose fiber is 2.32 cN/dtex and the twist is 80 T/10cm. The preparation of hyaluronic acid viscose fiber takes viscose as the substrate and adopts microcapsule technology to add hyaluronic acid into the matrix material under the protection of the shell. During the spinning process, the shell of the microcapsule melts at high temperatures. After spinning, the shell is washed off and removed. Therefore, the obtained fibers contain evenly distributed hyaluronic acid.

The hyaluronic acid viscose fiber with hyaluronic acid content of 10.4 mg/kg was purchased from Qianwei Ecological Textile Co., Ltd. (Shenzhen). The graphene viscose fiber was purchased from Gaoxi Technology Co., Ltd. (Hangzhou). The nylon used in the veil and lining yarn was purchased from Huading Nylon Co., Ltd. (Yiwu). The viscose fiber was purchased from Grace Group Co., Ltd. (Yibin). Finally, the spandex was purchased from DuPont. Additionally, the content of graphene in the graphene viscose fiber was 0.15%. The yarn specifications are shown in Table 1.

In this paper, the experimental sample scheme of hyaluronic acid seamless knitted fabric is established by using the full experimental method, and three kinds of structures, including weft plain stitch, 1 + 1 mock rib, and 1 + 3 mock rib, which are commonly used in seamless knitted structures in autumn and winter, are set up. The knitting machine used to complete sample weaving is SM8-TOP2S of Santoni, the gauge is E28, feeders are 16, and the diameter of the cylinder is 14 inches. The coursewise density (P_A_) and walewise density (P_B_) are shown in Table 2. The reason for setting the loop density in this way is to ensure that the tightness between samples is similar, which is very important for the subsequent experimental steps of coating the samples in the arm.

In order to explore the interweaving ratio of hyaluronic acid viscose fiber and graphene viscose fiber, and the effects of veil raw materials and fabric structure on fabric properties, the interweaving ratio of hyaluronic acid viscose fiber and graphene viscose fiber was set to 100:0, 75:25, 50:50, and 25:75. Through this design, the influence of hyaluronic acid viscose fiber on moisture retention performance is reflected. Furthermore, the samples of viscose and nylon were set as veil raw materials to compare the performance differences between fabrics containing hyaluronic acid and ordinary raw materials. The sample scheme design in this paper is shown in Table 3.

According to the established sample scheme, the weaving and proofing of 18 knitting samples were completed. For the needs of subsequent testing and analysis, the gram weight per square meter of all samples was tested, and the results are shown in Table 4.

### 2.2. Water Content Test

Skin moisture content is an important indicator of skin dryness, which is usually measured by the following three methods: the first, after the subjects are housed in an environment with a certain temperature and humidity for 20 min, uses a skin moisture tester for measurement [13,14]. The second method is according to the calculation of capacitance, and the skin water content is obtained indirectly [15]. Thirdly, the speed of skin surface water loss is reflected by measuring the trans-epidermal water loss rate of the skin, which refers to the difference in skin water content before and after a certain period of time. This method is often used to evaluate the effect of skincare products [16]. This subject adopts the first kind of stratum corneum water content detector (Corneometer CM825).

Principle: Based on the considerable change in the dielectric constant (<7) of water and other substances, according to the different water content, the appropriately shaped measuring capacitor will change with the change in the capacitance of the skin, and the capacitance of the skin is within the measurement range. Then, the water content of the skin is measured. The result is represented by the set MMA (0~150). In this way, because the tested area is effectively bonded with the test probe, there is essentially no small current passing through the tested area, and the test results are relatively accurate.

#### Skin Moisture Content Test Method

Test standard: Refer to QB/T 4256-2011 “Guidelines for evaluation of moisturizing efficacy of cosmetics” [17].

Experimental equipment: MPA6 multi-probe test system and corneometer cm825 skin moisture test probe.

Experimental environment and sample: The test environment temperature was 20 °C to 22 °C and the humidity is 40% to 60%. To reduce errors, the test environment was monitored in real time. The measuring area was marked 5 cm away from the palm on the inner side of the arm, and the test area was 3 cm × 3 cm; both left and right hands were marked. The sample was made to cover the whole forearm for one week. Velcro was pasted on the fabric surface to fix the fabric to cover the whole forearm. At the same time, it was ensured that only a single layer of fabric was covered on the marked side. The marking and sample coating status are shown in Figure 1 and Figure 2.

Subjects: There were 20 effective healthy subjects in Zhejiang, including 10 females and 10 males, aged 22–25 years.

Subject conditions: Skin moisture was measured in the forearm test area of the subject, and the measured basic value was between 15 and 45 (corneometer unit, c.u.). The subjects were required to have no serious systemic diseases, immune deficiencies, or autoimmune diseases, no skin treatment, beauty, or other factors that may have affected the results, no highly sensitive physique, and must not have used hormone drugs or immunosuppressive agents in the past month.

Experimental procedures: The skin was required to avoid contact with water 1~3 h before the test. Before the formal test, the subjects entered a room that met the standards, exposed their forearms to the environment, placed them flat on the table, with the test part facing up, remained relaxed, sat still for 30 min, and did not drink any liquid during the test.

Pre-experiment: After the fabric was in contact with the skin for a period of time, the water content of the skin reached a new balance. Therefore, before the formal test, a pre-experiment had to be carried out to determine the time when the skin reached the balance state. We conducted the test after debugging and calibrating the instrument according to the operating instructions of the skin moisture tester. During the test, the corneometer cm825 moisture test probe was placed at a 90° right angle to the forearm of the human body; then, we used an appropriate certain pressure to press on the surface of the tested skin. The system to which the probe was connected displayed the test results. We randomly selected test parts in the marked area, tested each part five times, and took the average value as the initial value. After the initial value test, the fabric was wrapped around the forearm, and the skin moisture test was carried out for 15 min, 30 min, and 1 h. It was found that the test results of covering the forearm for 30 min and 1 h were essentially consistent, which means that the skin moisture reached the equilibrium state. Therefore, the covering time of the fabric was determined as 30 min.

Formal test: After the subjects entered the test environment, we marked both the left and right arms according to the experimental requirements. The subjects were asked to sit still for 30 min, and then the initial value *T*_0_ (before covering the fabric sample) was measured. After this, we wrapped the fabric around the forearm and ensured that the fabric in the marked area was in a single-layer state. When the subjects sat still for another 30 min, they were tested to obtain *T*_1_. The test state is shown in Figure 3.

### 2.3. Measurement of Skin Trans-Epidermal Water Loss

The water of the human body evaporates continuously through the skin, which is an important part of human metabolism. Transdermal water loss (TEWL) is described in g/h/m^2^ (per hour per square meter). As long as the skin barrier layer is slightly damaged, even with very small damage that cannot be observed by the naked eye, the TEWL value of percutaneous water shunt loss will also increase.

Principle: We adopted Fick’s diffusion law- dm/dt = D·A·dp/dx, where A is the area (m^2^), D is the diffusion constant (0.0877 g/m.g.mmHg), M is the diffusion of water (g), T is the time (h), P is the steam pressure (mmHg), and X is the distance from the measuring point on the skin surface (m). The measuring probe was composed of two groups of temperature and humidity sensors. Its shape and size could prevent the influence of air flow on the measured data and it could be calibrated. By measuring the difference between the skin moisture content at the beginning and end nodes of a period of time, the speed of water loss on the skin surface could be obtained [18].

#### Test Method for Trans-Epidermal Water Loss

Test standard: Refer to QB/T 4256-2011 “Guidelines for evaluation of moisturizing efficacy of cosmetics” [17].

Experimental equipment: MPA6 multi-probe test system and Tewameter TM300 skin moisture loss test probe.

Experimental environment and sample: It was the same as in the skin moisture content test.

The subject and test conditions were the same as those described for the skin moisture content test.

Experimental procedures: After each skin moisture content test, the skin trans-epidermal water loss was tested directly. During the test, the tewameter TM300 skin moisture loss test probe was fitted with the human forearm at a 90° right angle. We clicked the start button, the instrument automatically collected the TEWL value per second, and the instrument automatically stopped the test after 30 s. After this, the software automatically drew a curve, including the deviation value and average value of TEWL. The test status is shown in Figure 4.

During the test, the testers were required to avoid exhaling above the test probe, and the surrounding environment was kept stable to prevent the gas in the external environment from affecting the results.

## 3. Results and Discussion

### 3.1. Test Results and Analysis of Skin Moisture Content

The corresponding skin water content of each sample was calculated according to Formula (1), and the average value was taken to obtain the test result of the skin water content change rate.
(1)$$\mathrm{Skin}\;\mathrm{moisture}\;\mathrm{content}\;\mathrm{change}\;\mathrm{rate}\%=\frac{T_{1}-T_{0}}{T_{0}}\times 100\%$$

The results of the average skin water content change rate of the human body are shown in Table 5.

Through SPSS and two-way ANOVA, it was found that the change of the fabric structure and the material in the fabric was *p* < 0.05. Therefore, these two factors have a significant impact on the change rate of human skin moisture content, and the change of material in fabric has a greater impact on the change rate of human skin moisture content. The inter-subject effect test is shown in Table 6, and the data conform to a normal distribution.

The Duncan method was used to compare the differences between fabric structure and raw material changes. The results are shown in Table 7 and Table 8. In terms of the fabric structure, the weft plain stitch fabric and 1 + 1 mock rib fabric accounted for the same subset, and 1 + 3 mock rib fabric accounted for a subset, indicating that there were significant differences between 1 + 3 mock rib fabric, 1 + 1 mock rib, and weft plain stitch fabric. Among them, the water content change rate of human skin corresponding to 1 + 1 mock rib fabric was the highest, and the moisturizing effect was the best, followed by weft plain stitch fabric and 1 + 3 mock rib fabric.

In the use of raw fabric materials, the data of nylon fabric and viscose fabric accounted for the same subset, indicating that there was no significant difference between the two. The interweaving ratio of hyaluronic acid viscose fiber/graphene viscose fiber changed to 75:25, 50:50, and 25:75, and they were in the same subset. However, the hyaluronic acid viscose fiber/graphene viscose fiber with an interweaving ratio of 100:0 accounted for a separate subset. This indicates that the data of nylon fabric, viscose fabric, and hyaluronic acid viscose fiber/graphene viscose fiber interweaving fabric were significantly different. When the interweaving ratio of hyaluronic acid viscose fiber and graphene viscose fiber was 100:0, the corresponding skin water content change rate of the fabric was the highest, and its moisturizing effect was also the best.

Results in Figure 5 show that when the fabric structure adopts 1 + 1 mock rib and the raw material adopts hyaluronic acid viscose fiber/graphene viscose fiber (100:0), the fabric has the best moisturizing effect on human skin.

### 3.2. Test Results and Analysis of Trans-Epidermal Water Loss

The corresponding transdermal water loss of each sample was calculated according to Formula (2), and the average value was taken to obtain the test result of the transdermal water loss change rate.
(2)$$\mathrm{Change}\;\mathrm{rate}\;\mathrm{of}\;\mathrm{water}\;\mathrm{loss}\;\mathrm{through}\;\mathrm{skin}\%=\frac{T_{1}-T_{0}}{T_{0}}\times 100\%$$

The results of the average trans-epidermal water loss change rate are shown in Table 9.

Through SPSS and two-way ANOVA, it was found that the change in fabric structure and the material in the fabric was *p* > 0.05, so these two factors had no significant effect on the change rate of human skin moisture content. From the data calculation results, it can be concluded that the change rate of water loss through the epidermis is not obvious and has no significant impact on the fabric structure and the raw materials used in the fabric. It may not be able to repair the skin barrier in a short time, or the content of hyaluronic acid in the fabric could not cause a change in this index.

## 4. Conclusions

In this paper, aiming at the symptoms of human skin dryness caused by environmental dryness in autumn and winter, 18 seamless knitted samples were woven incorporating the moisturizing function of hyaluronic acid viscose fiber. The structure of the samples adopted three kinds of seamless knitting structures most commonly used in autumn and winter. Then, the 18 knitted samples were used to test the skin water content and trans-epidermal water loss of 20 subjects. The test results were analyzed, and the following conclusions were drawn from the fabric structure and raw materials.

1. The surfaces of 1 + 3 mock rib and 1 + 1 mock rib fabrics have convex structures, while the bulge formed by 1 + 1 mock rib is smaller, which makes the area where hyaluronic acid viscose fiber can contact the skin larger. Therefore, hyaluronic acid can play a better moisturizing role in the structure of 1 + 1 mock rib.

2. Compared with weft plain stitch fabric, 1 + 1 mock rib fabric is thicker and has stronger thermal insulation performance, so the heat loss of the human body is slower, and the skin can sweat more easily, so as to form a water film between the surface of the human skin and the fabric, which has the effect of water retention.

3. In terms of structure, the order of the influence of fabric on the moisturizing performance of human skin from large to small is 1 + 1 mock rib fabric > weft plain stitch fabric > 1 + 3 mock rib fabric.

4. Hyaluronic acid has a good moisturizing effect on the skin. Water molecules are locked in its double-helix columnar structure through the hydrogen bonds in hyaluronic acid molecules, so that water is not easily lost. Therefore, when the interweaving ratio of hyaluronic acid viscose fiber is reduced, the moisturizing effect of its fabric on the skin is also reduced.

5. Compared with nylon fabric, viscose fabric has good moisture absorption and a better moisturizing effect on human skin.

6. In terms of the use of raw materials, the order of the influence of fabrics on the moisturizing performance of human skin is hyaluronic acid viscose fiber/graphene viscose fiber fabric (100:0 > 75:25 > 50:50 > 25:75) > viscose fabric > nylon fabric.

7. In 18 samples, the fabric formed by 1 + 1 mock rib and hyaluronic acid viscose fiber/graphene viscose fiber (100:0) had the best moisturizing effect on skin. In the experiment of skin trans-epidermal water loss, the structure and raw materials used in the fabric both had little effect on the test results.

## Figures and Tables

**Figure 1 materials-15-01806-f001:**
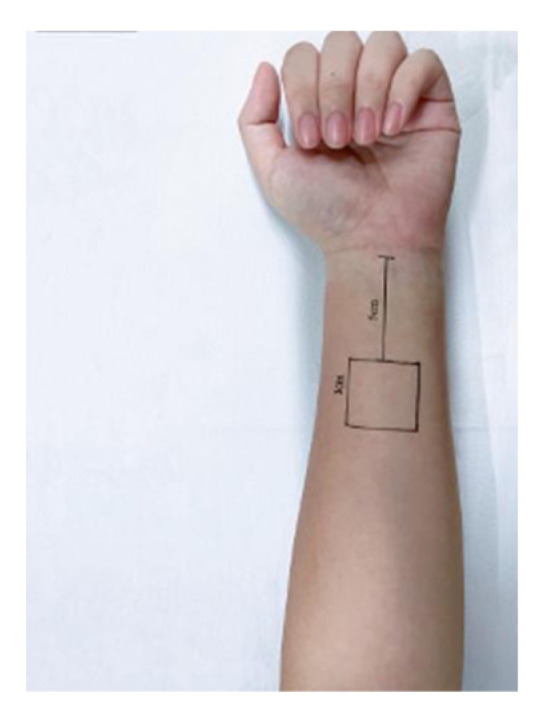
Schematic diagram of test area marking.

**Figure 2 materials-15-01806-f002:**
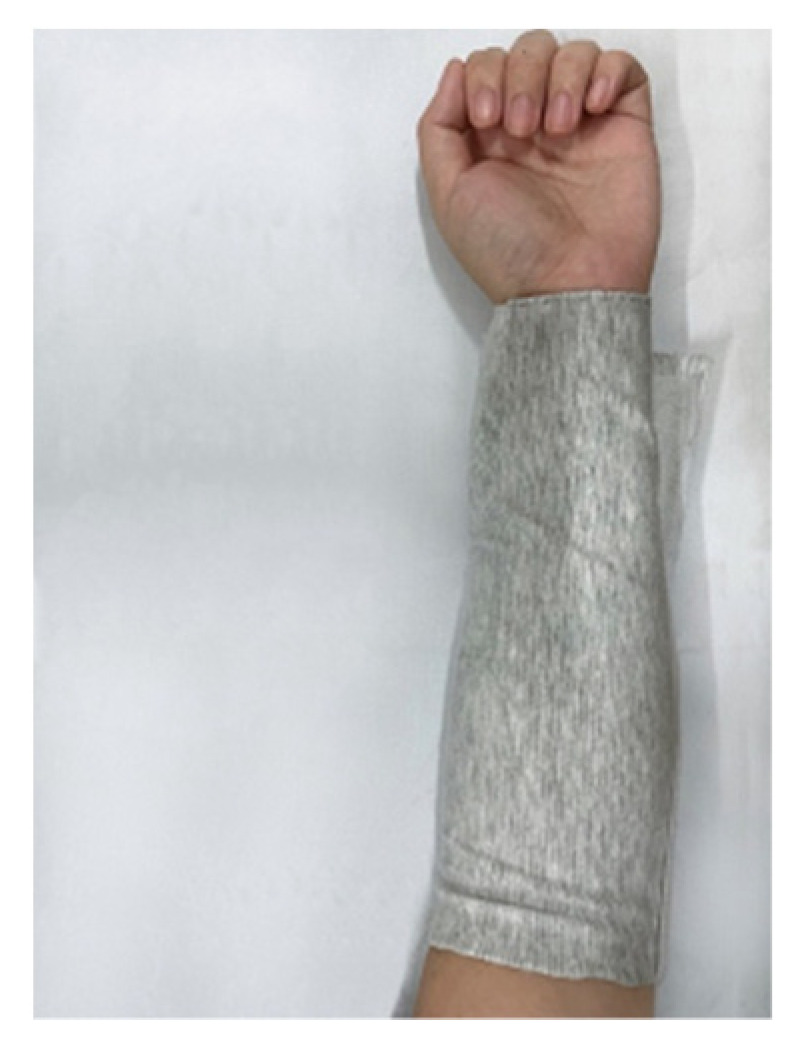
Sample wrapping arm diagram.

**Figure 3 materials-15-01806-f003:**
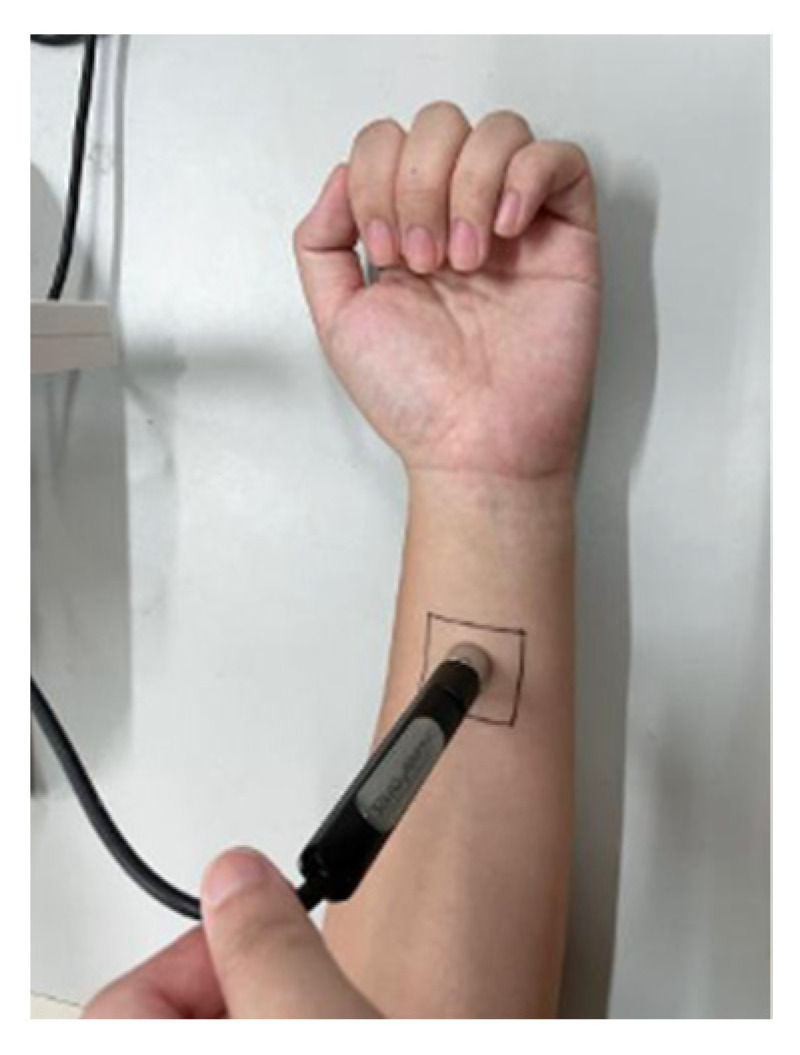
Test diagram of corneometer cm825 moisture test probe.

**Figure 4 materials-15-01806-f004:**
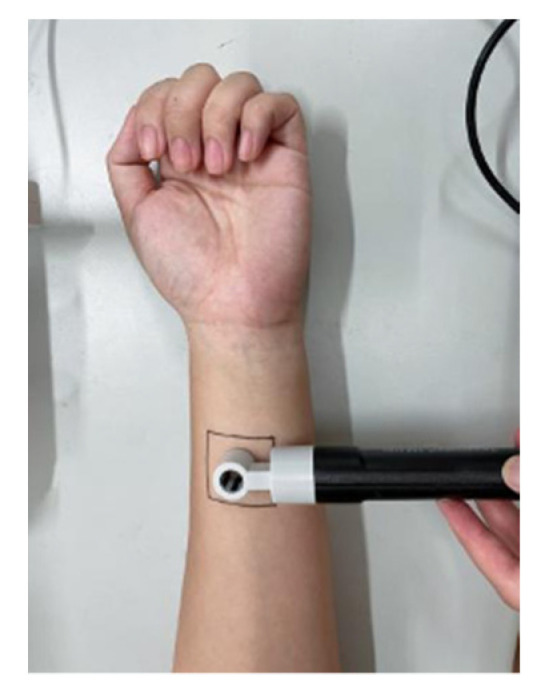
Tewameter TM300 test diagram of skin moisture loss test probe.

**Figure 5 materials-15-01806-f005:**
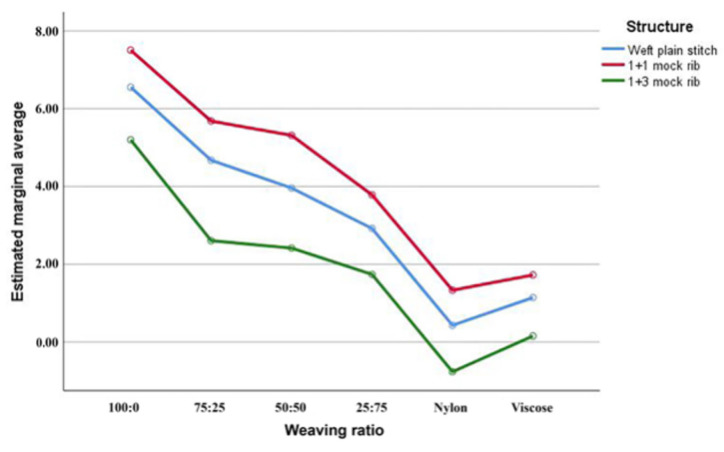
Test results of skin moisture content change rate.

**Table 1 materials-15-01806-t001:** Yarn specifications.

Yarn Type	Fineness	Filaments
Hyaluronic acid viscose fiber	105D	36F
Graphene viscose fiber	105D	36F
Nylon	100D	48F
Viscose	105D	48F
Nylon/spandex-coated yarn	20D/70D	6F/1F

**Table 2 materials-15-01806-t002:** Stitch density of fabric samples.

Fabric Structure	P_A_/(Wale/5 cm)	P_B_/(Wale/5 cm)
Weft plain stitch	55	150
1 + 1 mock rib	55	180
1 + 3 mock rib	55	200

**Table 3 materials-15-01806-t003:** The fabric sample scheme.

Sample Numbers	Veil Raw Materials	Weaving Ratio	Fabric Structure
1	Hyaluronic acid viscose fiber/Graphene viscose fiber	100:0	Weft plain stitch
2	Hyaluronic acid viscose fiber/Graphene viscose fiber	75:25	Weft plain stitch
3	Hyaluronic acid viscose fiber/Graphene viscose fiber	50:50	Weft plain stitch
4	Hyaluronic acid viscose fiber/Graphene viscose fiber	25:75	Weft plain stitch
5	Hyaluronic acid viscose fiber/Graphene viscose fiber	100:0	1 + 1 mock rib
6	Hyaluronic acid viscose fiber/Graphene viscose fiber	75:25	1 + 1 mock rib
7	Hyaluronic acid viscose fiber/Graphene viscose fiber	50:50	1 + 1 mock rib
8	Hyaluronic acid viscose fiber/Graphene viscose fiber	25:75	1 + 1 mock rib
9	Hyaluronic acid viscose fiber/Graphene viscose fiber	100:0	1 + 3 mock rib
10	Hyaluronic acid viscose fiber/Graphene viscose fiber	75:25	1 + 3 mock rib
11	Hyaluronic acid viscose fiber/Graphene viscose fiber	50:50	1 + 3 mock rib
12	Hyaluronic acid viscose fiber/Graphene viscose fiber	25:75	1 + 3 mock rib
13	Nylon		Weft plain stitch
14	Nylon		1 + 1 mock rib
15	Nylon		1 + 3 mock rib
16	Viscose		Weft plain stitch
17	Viscose		1 + 1 mock rib
18	Viscose		1 + 3 mock rib

**Table 4 materials-15-01806-t004:** Gram weight per square meter of samples.

Sample Numbers	GSM/(g/m^2^)	Sample Numbers	GSM/(g/m^2^)
1	300.93	10	467.70
2	311.57	11	457.86
3	305.91	12	465.29
4	306.21	13	252.25
5	368.20	14	297.44
6	367.82	15	369.11
7	367.83	16	293.18
8	370.01	17	370.68
9	460.85	18	465.20

**Table 5 materials-15-01806-t005:** Skin moisture content change rate.

Sample Numbers	Veil Raw Materials	Weaving Ratio	Skin Moisture Content Change Rate (%)
1	Hyaluronic acid viscose fiber/Graphene viscose fiber	100:0	6.55
2	Hyaluronic acid viscose fiber/Graphene viscose fiber	75:25	4.68
3	Hyaluronic acid viscose fiber/Graphene viscose fiber	50:50	3.96
4	Hyaluronic acid viscose fiber/Graphene viscose fiber	25:75	2.92
5	Hyaluronic acid viscose fiber/Graphene viscose fiber	100:0	7.51
6	Hyaluronic acid viscose fiber/Graphene viscose fiber	75:25	5.68
7	Hyaluronic acid viscose fiber/Graphene viscose fiber	50:50	5.31
8	Hyaluronic acid viscose fiber/Graphene viscose fiber	25:75	3.78
9	Hyaluronic acid viscose fiber/Graphene viscose fiber	100:0	5.20
10	Hyaluronic acid viscose fiber/Graphene viscose fiber	75:25	2.60
11	Hyaluronic acid viscose fiber/Graphene viscose fiber	50:50	2.41
12	Hyaluronic acid viscose fiber/Graphene viscose fiber	25:75	1.74
13	Nylon		0.43
14	Nylon		1.33
15	Nylon		−0.77
16	Viscose		1.14
17	Viscose		1.72
18	Viscose		0.15

**Table 6 materials-15-01806-t006:** Variance analysis of skin water content change rate.

Inter-Subject Effect Test					
Dependent Variable: Skin Moisture Content Change Rate					
Source	Class III Sum of Squares	Freedom	Mean Square	F	Significance
Modified model	932.093 a	17	54.829	4.816	0.000
Intercept	1764.444	1	1764.444	154.974	0.000
Structure	165.204	2	82.602	7.255	0.001
Materials	758.110	5	151.622	13.317	0.000
Structure* Materials	8.779	10	0.878	0.077	1.000
Error	1844.435	162	11.385		
Total	4540.972	180			
Corrected total	2776.528	179			

a. R square = 0.336 (After adjustment R square = 0.266).

**Table 7 materials-15-01806-t007:** Duncan method results for fabric structure sample difference significance analysis with skin moisture content change rate.

Structure	Number of Cases	Subset	
		1	2
1 + 3 mock rib	60	1.8903	
Weft plain stitch	60		3.2795
1 + 1 mock rib	60		4.2228
Significance		1.000	0.128
The error term is the mean square (error) = 11.385.			
Sample size using harmonic mean = 60.000.			
Alpha = 0.05.			

**Table 8 materials-15-01806-t008:** Significance analysis of the difference between Duncan fabric raw material samples and the change rate of skin water content.

Materials	Number of Cases	Subset		
		1	2	3
Nylon	30	0.3317		
Viscose	30	1.0053		
25:75	30		2.8127	
50:50	30		3.8960	
75:25	30		4.3200	
100:0	30			6.4197
Significance		0.441	0.104	1.000
The error term is the mean square (error)= 11.385.				
Sample size using harmonic mean = 30.000.				
Alpha = 0.05.				

**Table 9 materials-15-01806-t009:** Change rate of trans-epidermal water loss.

Sample Numbers	Veil Raw Materials	Weaving Ratio	Change Rate of Trans-Epidermal Water Loss (%)
1	Hyaluronic acid viscose fiber/Graphene viscose fiber	100:0	1.85
2	Hyaluronic acid viscose fiber/Graphene viscose fiber	75:25	2.75
3	Hyaluronic acid viscose fiber/Graphene viscose fiber	50:50	3.43
4	Hyaluronic acid viscose fiber/Graphene viscose fiber	25:75	2.79
5	Hyaluronic acid viscose fiber/Graphene viscose fiber	100:0	3.70
6	Hyaluronic acid viscose fiber/Graphene viscose fiber	75:25	2.31
7	Hyaluronic acid viscose fiber/Graphene viscose fiber	50:50	4.48
8	Hyaluronic acid viscose fiber/Graphene viscose fiber	25:75	2.18
9	Hyaluronic acid viscose fiber/Graphene viscose fiber	100:0	2.39
10	Hyaluronic acid viscose fiber/Graphene viscose fiber	75:25	5.17
11	Hyaluronic acid viscose fiber/Graphene viscose fiber	50:50	3.12
12	Hyaluronic acid viscose fiber/Graphene viscose fiber	25:75	2.75
13	Nylon		−0.11
14	Nylon		1.53
15	Nylon		3.27
16	Viscose		1.99
17	Viscose		3.23
18	Viscose		1.21

## Data Availability

The data are not publicly available.

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
