# Peer review of "Study on the Structure and Skin Moisturizing Properties of Hyaluronic Acid Viscose Fiber Seamless Knitted Fabric for Autumn and Winter"

_materials, 2022, doi:10.3390/ma15051806_

Round 1

Reviewer 1 Report

What is amount of graphene in graphene viscose fibre?

What is staple length, crimp, denier of all fibres used to develop all the yarns mentioned in page 2, line 60-65?

In page2, from line 60-65, the right way to represent is staple yarn/filament yarn composed of viscose fiber or graphene viscose fibre etc? How many filaments in 100D nylon yarn?

How you made hyaluronic acid viscose fiber?

It is must to include yarn characteristics such strength, tpi if staple spun yarn etc?

If mentioned yarns were procured, provide details of supplier etc.

It is necessary to rewrite materials section with all required details in clear way.

It is must to provide knitted fabric details, such as structure, gsm, thickness loop density etc.

Authors are requested to provide Knitting machine details such as gauge, feeders, diameter of cylinder?

Author Response

Thank you for your precious comments and advice. Those comments are all valuable and very helpful for revising and improving our paper, as well as the important guiding significance to our researches. We have studied comments carefully and have made correction which we hope meet with approval. Revised portion are marked in red in the paper. Please see the attachment about the main corrections in the paper and the responds to the reviewer’s comments.

Reviewer 2 Report

In the manuscript (materials-1534153), the authors studied the effects of fabric structure and raw materials on moisture content of the skin. However, there is not sufficient background information and sufficient literature examples in the introduction, the introduction section can be improved with recent examples. The citations throughout the manuscript can not be viewed, there is an error "Error! Reference source not found.." Conversely, the "Materials and Methods" section is too long and difficult to follow. The language used in explaining the methods sounds like a user's guide. As an example, in line 109 "Mark the measuring area 5cm away from the palm on the inner side of the 109 arm, and the test area is 3cm × 3cm, mark with both left and right hands." should be changed to "The measuring area was marked 5 cm away from the palm...." The number of figures is too high for a materials and methods section, for instance the figures of the apparatus are not necessary.  However, there are not enough figures in results and discussion part. Results and discussion section is shorter than materials and methods section, which is very uncommon. The results given are presented as tables, which is not easy compare, i would prefer graphical presentation so that readers can compare the differences based on fabric type and structure. I would also prefer to see more visual data rather than just numbers, which would increase the quality of the paper, such as imaging the epidermis with a microscope, because the authors defend that these fabrics reduce skin dryness. In Table 6, it must be clearly stated what are 25:75, 50:50, 75:25 and 100:0, at least in the caption. In Figure 7, the x axis is not clearly presented enough. In line 243, the authors claimed that 1+1 mock rib fabric showed stronger thermal insulation performance, how was this measured? The quality of writing should also be improved in results and discussions. There are some very long sentences in the text, it can be a good idea to divide them in two sentences which would be easier to understand for the readers (such as line 225-230, 279-282). I am glad to reconsider the revised version then.

Author Response

(The authors gave the same response as above.)

Round 2

Reviewer 1 Report

In Table1, it is good to include number of filaments along with yarn denier for all samples.

In Table 4, it should be sample numbers, instead of number of samples. Same changes should be considered for entire text and tables 3, 5 etc..

Author Response

Thank you for your precious comments and advice. Each comment and suggestion were helpful. We have revised the manuscript according to your detailed suggestions. The revised manuscript with revision marks were attached as the supplemental materials and for easy check/editing purpose. Please see the attachment.

Reviewer 2 Report

I thank the authors for the modifications in the manuscript. However, i still think the introduction is insufficient. There are not sufficient background from recent works on the use of hyaluronic acid in textiles and moisture content of textiles. The most recent examples given by the authors are from 2009-2017, but there are more recent studies which should be cited. The novelty of this paper is not clearly described. The results presented are still confusing, i suggest the authors to divide veil raw materials and weaving ratio into two seperate columns in Table 3, 5 and 9. The x axis of the graph (Figure 5) is still missing and the values on x axis are not corrected. The results are disorganized, for example i think line 381-384 should come after line 354-355. The flow of information is broken. "From the fabric structure" and "from the raw materials" subtitles can be removed from the results and discussion part for a more continuous flow. This point of view can be added to the conclusions, the results can be summarized from the point of the fabric structure and raw materials in the conclusion part. I still think some other type of data would contribute to the manuscript.

Author Response

(The authors gave the same response as above.)
